# Host Specificity of the Parasitic Wasp *Anaphes flavipes* (Hymenoptera: Mymaridae) and a New Defence in Its Hosts (Coleoptera: Chrysomelidae: *Oulema* spp.)

**DOI:** 10.3390/insects11030175

**Published:** 2020-03-10

**Authors:** Alena Samková, Jiří Hadrava, Jiří Skuhrovec, Petr Janšta

**Affiliations:** 1Department of Plant Protection, Faculty of Agrobiology, Food and Natural Resources, Czech University of Life Sciences Prague, Kamýcká 129, CZ-165 00 Prague 6—Suchdol, Czech Republic; 2Department of Zoology, Faculty of Science, Charles University, Viničná 7, CZ-128 43 Prague 2, Czech Republic; HadravaJirka@seznam.cz (J.H.); petr.jansta@natur.cuni.cz (P.J.); 3Institute of Entomology, Biological Centre, Czech Academy of Sciences, Branišovská 31, CZ-370 05 České Budějovice, Czech Republic; 4Group Function of Invertebrate and Plant Biodiversity in Agro-Ecosystems, Crop Research Institute, Drnovská 507, CZ-161 06 Praha 6—Ruzyně, Czech Republic; jirislavskuhrovec@gmail.com

**Keywords:** parasitoid-host interaction, biological control, host spectrum, Mymaridae

## Abstract

The parasitic wasp *Anaphes flavipes* (Förster, 1841) (Hymenoptera: Mymaridae) is an important egg parasitoid of cereal leaf beetles. Some species of cereal leaf beetle co-occur in the same localities, but the host specificity of the wasp to these crop pests has not yet been examined in detail. A lack of knowledge of host specificity can have a negative effect on the use of this wasps in biological control programs addressed to specific pest species or genus. In this study, laboratory experiments were conducted to assess the host specificity of *A. flavipes* for three species of cereal leaf beetles (*Oulema duftschmidi* Redtenbacher, 1874, *Oulema gallaeciana* Heyden, 1879 and *Oulema melanopus* Linnaeus, 1758) in central Europe. For the first time, a new host defence against egg parasitoids occurring in *O. gallaeciana* from localities in the Czech Republic, a strong dark sticky layer on the egg surface, was found and described. The host specificity of *A. flavipes* was studied in the locality with the presence of this defence on *O. gallaeciana* eggs (the dark sticky layer) (Czech Republic) and in a control locality (Germany), where no such host defence was observed. Contrary to the idea that a host defence mechanism can change the host specificity of parasitoids, the wasps from these two localities did not display any differences in that. Respectively, even though it has been observed that eggs with sticky dark layer can prevent parasitization, the overall rate of parasitization of the three species of cereal beetles has not been affected. However, in our view, new host defence can influence the effects of biological control, as eggs of all *Oulema* spp. in the locality are protected against parasitization from the wasps stuck on the sticky layer of the host eggs of *O. gallaeciana*.

## 1. Introduction

Parasitic wasps occur in high numbers, both in terms of species diversity and absolute numbers of individuals [1]. Despite differing estimates, the diversity of parasitic wasps is assumed to be over one million species, with roughly every tenth species of insects being a parasitic hymenopteran [1,2]. Parasitic wasps attack a wide range of hosts, and they play an important role in the biodiversity and balance of natural ecosystems and agriculture [3,4]. Especially in agriculture, parasitic wasps can be used to reduce important pest insects in biological control programmes [5]. Biological control can be implemented either as modification of landscaping strategies that support natural enemies (conservation biological control) [6] or as releasing (once or repeatedly) of parasitoids or predators in the infested fields or areas (augmentative control) [7,8]. This latter strategy is effective for organic farming and natural protected areas, or if the pests acquire resistance to chemical spraying [9,10,11,12]. For example, for augmentative biological control, 170 species of parasitoids are used only in Europe [8]. In some cases, the effectiveness of biological control is up to 100%, e.g., the wasp *Cosmocomoidea ashmeadi* (Girault, 1915) (Hymenoptera: Mymaridae) controlling the leafhopper *Homalodisca coagulata* (Say, 1832) (Hemiptera: Cicadellidae) [13], or *Anaphes nitens* (Girault, 1928) (Hymenoptera: Mymaridae) against the weevil *Gonipterus scutellatus* Gyllenhal, 1833 (Coleoptera: Curculionidae) [14,15]. Many parasitoids do not have 100% efficiency, and therefore, methods for improving the effectiveness of natural enemies or biological control have been sought [16].

The efficiency of biological control can be increased with detailed knowledge of the host specificity of parasitoids, and host specificity is one of the primary criteria for evaluating the risks of biological control organisms to nontarget organisms [17]. The host range is generally characterized as the set of species on which a control organism can feed and develop [17], all organisms in a given habitat are potentially a host for parasitoids, but their quality and parasitoid approbation is different [1]. The host may be attacked by one to over twenty parasitoids at a time, and the most vulnerable hosts to parasitization appear to be herbivorous insects [18]. Successful parasitization requires the parasitoid first to locate the host’s habitat, then perform a specific behavioral routine to finally lay eggs on or into the host [19]. Parasitoids are also capable of learning novel signals that improve their search efficiency [1]. The hosts are not merely passive participants in this process [3]. They have mechanical, physiological and immune defences against parasitoids and are in a constant evolutionary arms race; whatever defensive mechanisms the host invents, the parasitoid tries to overcome [20,21].

Our study is focused on the host specificity of the potential biological agent, *Anaphes flavipes* (Föster, 1841) (Hymenoptera: Mymaridae). The host spectrum of this wasp includes the rare *Lema* spp. and the widespread *Oulema* spp. (*O. duftschmidi* Redtenbacher, 1874, *O. gallaeciana* Heyden, 1879, and *O. melanopus* Linnaeus, 1758) [22,23]. Larvae and adults of *Oulema* species damage the leaves of cereals (barley, wheat and oats), and they are an economically important crop pest in Europe and North America [24,25,26]. For example, in agricultural areas around the world, insect pests reduce grain crop yields by 5% to 20 % every year [27]. The use of parasitic wasps for biological control has been repeatedly tested [28,29,30]. In this context, the host spectrum of *A. flavipes* was examined for six taxons, *Crioceris duodecimpunctata* (Linnaeus, 1758); *Oulema sayi* (Crotch, 1873); *Lema nigrovittata* (Guérin-Méneville, 1844); *Lema daturaphila* Kogan and Goeden, 1970 (as *L. trilineata* (Olivier, 1808)); *Lema trilineata californica* (Schaefer Krauss 1947) and *Lema trivittata trivittata* Say, 1824) by Maltby et al. [31]; however, a current common host species, *O. duftschmidi*, was not included because, until 1989, it was assigned to *O. melanopus* [32]. In relation to the study of host specificity of *A. flavipes*, we describe a new type of host defense against egg parasitoid as a dark sticky layer on host eggs of *O. gallaeciana* in Czech localities. This sticky layer can completely prevent parasitization, because any females adhere to the sticky layer and are unable to either parasitize or release herself. The eggs defense of cereal beetles against parasitization by parasitoids has been proposed so far only by Anderson and Paschke [22] as a strong selective pressure on the rapid development of beetle larvae. The wasps reject host eggs older than 72 h, probably because the forming larvae could damage the parasitoid egg by sclerotized mandibles. However, there is no experimental evidence for this claim.

First, laboratory experiments were carried out to assess the host specificity of *Anaphes flavipes* for three widespread *Oulema* species (*O. duftschmidi*, *O. gallaeciana* and *O. melanopus*). Secondly, we described a new type of host defence against parasitoids on eggs and compared the host specificity of the wasp *A. flavipes* between metapopulations with or without the observed egg defence on three co-occurring *Oulema* species. The main aim was to test whether the wasp’s choice of host species can be affected by the presence or absence of the host defence.

## 2. Materials and Methods

### 2.1. Parasitic Wasps

Parasitic wasps (*A. flavipes*) were collected from host eggs of species *O. duftschmidi*, *O. gallaeciana* and *O. melanopus* in periods from the end of April until the end of June 2015–2016 in cereal fields in one locality in the Czech Republic (50.1385 N, 14.3695 E) and one locality in Germany (50.7787 N, 6.0381 E). The parasitized host eggs were stored in Petri dishes with moistened filter paper until the emergence of the adult wasps. These “wild” wasps were used as an initial population for rearing the next generations of parasitoids in an environmental chamber with conditions of 22 ± 2 °C, relative humidity of 40%–60% and 24 h light. All these “next generation” females and males of *A. flavipes* were bred in laboratory on the eggs *Oulema* species (*O. duftschmidi* and *O. melanopus*) and those used for experiments were at most 24 h old (post emergence). Each emerged female used in the experiment was immediately mated. Each mated female was then placed in a Petri dish with 12 host eggs (8 eggs of *O. duftschmidi* + *O. melanopus*, 4 eggs of *O. gallaeciana*). Before starting and during the experiments, the females were not fed, and they had constant access to water.

### 2.2. Host Species

The host species of the genus *Oulema* (*O. duftschmidi*, *O. gallaeciana* and *O. melanopus*) were obtained from the adults collected in localities in the Czech Republic (one at the same location as that used for parasitic wasps and one more in Police nad Metují (50.5277 N, 16.2456 E)) and Germany (near the city of Aachen (50.7763 N, 6.0838 E)). The hosts were collected using a net or by hand collection.

The *Oulema* species were divided into two groups: (1) *O. gallaeciana* (*Og*), and (2) *O. duftschmidi* (*Od*) + *O. melanopus* (*Om*). Although, *O. gallaecina* can be easily determined to species level using external morphological characters (such as body color), *O. duftschmidi* and *O. melanopus* are distinguishable only when the morphology of the genitals is applied, see [32]). Therefore each female of *Od* and *Om* which eggs have been used in experiments, were stored in ethanol and determined to species by genital preparation to enable assign host exact host species to each host egg used in the experiment. The *Oulema* species were bred in Petri dishes (diameter 8.5 cm, for pairs of hosts – *Od* + *Om*) or plastic boxes (10 × 10 × 5.5 cm or 20 × 20 × 18 cm, for more individuals – *Og*) with moistened filter paper in an environmental chamber at 22 ± 2°C and a relative humidity of 40%–60%. The beetles were fed with grain leaves and had unlimited access to water. Cereal leaf beetle lay eggs on the fresh leaves of cereals. In our experiment, every 24 h, the leaves of cereals were removed from Petri dishes and plastic boxes, and fresh leaves were given. The leaves with eggs were cut into pieces which contain only one host egg (approximately 1 cm long piece of leave), each piece was numbered and placed into Petri dish (up a total of 12 eggs—4 *Og* and 8 *Od* + *Om*) for parasitation by one wasp.

### 2.3. Laboratory Experiments

All laboratory experiments were performed in Petri dishes (diameter 8.5 cm) in a thermal cabinet at 22 ± 2 °C. Eggs were removed on the 9th or 10th day after parasitization, placed in 1.5 mL Eppendorf tubes and stored at the same temperature in the thermal cabinet. After wasps’ emergence, the number of parasitized host eggs by one female in relation to host species were measured.

#### 2.3.1. Host Defence

In 2012, at localities in the Czech Republic (50.5277 N, 16.2456 E; 50.1385 N, 14.3695 E), host eggs coated with a strong dark sticky layer (Figure 1A,B) were observed for the first time. The parasitization of eggs with and without the dark sticky layer was documented by photography using a Canon EOS70D camera equipped with a Canon MP-E 65/2.8 MACRO lens.

In 2016, *Oulema* adults from these two localities (50.5277 N, 16.2456 E; 50.1385 N, 14.3695 E) were collected, and all individuals were put into Petri dishes (one female and one male in one Petri dish or one female in one Petri dish) and divided into two groups: (1) *O. gallaeciana* (*Og*) (*n_Og_* = 82) and (2) *O. duftschmidi* (*Od*) + *O. melanopus* (*Om*) (*n_Od+Om_* = 100). Every 24 h for 15 d, the number of eggs laid (*n_Og_* = 1083, *n_Od+Om_* = 3280) by one female of *Oulema* was measured in three categories: (1) sticky dark (distinctive dark sticky layer; Figure 1A,B); (2) slightly sticky (yellow colour; the structure of the egg surface is not visible through the sticky layer; Figure 1C,D); (3) non-sticky (yellow colour, the structure of the egg surface is visible; Figure 1E,F) (Appendix A). The host eggs were photographed with the Canon EOS70D camera (*n_Og_* = 500, *n_Od+Om_* = 500). The differences in the proportion of eggs with the dark sticky layer between host species were tested by chi-square test.

#### 2.3.2. Host Specificity

Each female of *A. flavipes* (*n* = 59) from the two localities with *Og* eggs with a sticky layer (Czech Republic; see Appendix A for details) had 12 host eggs (4× *Og* with a sticky layer and 8× *Od* + *Om* without a sticky layer) available for parasitization for 8 h in a Petri dish. The host specificity of these wasps was compared to that of the wasps from control German locality where no sticky layer on *Og* eggs was observed (see Appendix A for details). Each female of *A. flavipes* (*n* = 18) from German locality had 12 host eggs (4× *Og* and 8× *Od* + *Om*) available for parasitization for 8 h in a Petri dish. The host beetles *Om* + *Od* (after laying host eggs) were stored in 96% ethanol and identified to species by their genitalia (see *Host species*). For each female, the total number of parasitized host eggs of the three host species (*Od*, *Og*, *Om*) was measured (Appendix A). The wasps that stuck to the sticky surface of the eggs were discarded from the experiment.

Control locality in Germany: The *Og* females and males from Germany were collected at the same time as the parasitic wasps from the localities of cereals near the city of Aachen (50.7763 N, 6.0838 E). One female and one male or only one female were placed in Petri dishes with moistened filter paper and crop leaves. Every 24 h, the leaves of cereals with host eggs were removed from each Petri dish and were replaced by fresh leaves and water. All obtained host eggs (*n* = 45) were photographed using a Canon EOS70D camera and were recorded category of sticky layer (see Figure 1). The occurrence of the dark sticky layer on their eggs (Figure 1A,B) was not observed.

In order to test the negative effect of sticky layer on the host specificity of *A. flavipes*, for each *A. flavipes* female, the preference for *Od* + *Om* vs. *Og* was tested by a binomial test. In this case, a one-sided test was used due to the negative effect of the dark sticky layer on the parasitization of *Og* eggs. Fisher’s method of meta analysis was used for joining the *p*-values (function “sumlog” in R package “metap” was used for this purpose). Females from the Czech Republic and Germany were tested separately due to differences in the presence of the dark sticky layer in their native ecosystems. R version 3.3.3 [33] was used for all statistical analyses.

## 3. Results

### 3.1. Host Defence

The eggs of *Og* (*n* = 82) and *Od* + *Om* (*n* = 100) females were classified into the three categories (1) sticky dark eggs (distinctive dark color; sticky layer); (2) little sticky eggs (yellow color; the structure of the egg surface is not visible under the sticky surface) and (3) non-sticky eggs (yellow color; the structure of the egg surface is visible, Figure 1) for the locality in the Czech Republic. The eggs with a dark sticky layer (Figure 1A,B) were significantly more prevalent in the *Og* species (Figure 2A) than in *Om* and *Od* (Figure 2B) (X-squared test, *n*_eggs*Og*_ = 1083, *n*_eggs*Od*+*Om*_ = 3280, X-squared = 2857.5, df = 2, *p* < 0.001).

The behavior that precedes parasitization and the parasitization itself is documented in Figure 3 and Figure 4, which show the parasitization of the eggs with a sticky layer. Three behavioral situations during the parasitization of *Og* eggs with a dark sticky layer (Figure 1A,B) from the Czech locality were observed and described:(1)The female adheres to the sticky layer and is unable to either parasitize or release herself;(2)The wasp is able to parasitize the egg but cannot release herself from the egg surface;(3)The wasp successfully parasitizes the eggs and leaves the host, but afterwards, she must clean herself.

### 3.2. Host Specificity

Change of the host specificity of *A. flavipes* wasps from localities with host defense presented as a dark sticky layer on the host eggs of *Og* was not statistically confirmed (*p* = 0.99, *p* = 0.60, *n* = 59, respectively; meta-analysis on *p*-values from binomial tests, Figure 5A). The German locality was used as a negative control due to the absence of *Og* eggs with a sticky layer (*n* = 18, Figure 5B).

## 4. Discussion

The parasitic wasp *A. flavipes* was introduced in the 1970s from Europe (France, Germany, Italy) to the USA [22,28], and the host spectrum of these wasps was examined for six species [31]. However, a common current host species, *O. duftschmidi*, was not included in the previous study because until 1989, it was assigned to *O. melanopus* [32]. However, *O. duftschmidi* occurs together with other crop beetles in the same grain field agroecosystems [34]. The host specificity of wasps for three common species of crop beetles (*O. duftschmidi*, *O. gallaeciana* or *O. melanopus*) was reviewed in central Europe. Our experiments found that the *A. flavipes* populations from central Europe do not show host specificity due to the absence of any female preference for specific hosts of the genus *Oulema*. This confirmation may be partly due to the fact that the choice of a host normally depends on phylogeny and host ecology [35], which are both extremely similar among our three hosts. However, the finding that wasps parasitize all three *Oulema* species without substantial preference is extremely beneficial for their practical use in biological control programmes.

The effectiveness of biological control using parasitoids may be strongly affected by host defence. Many insect species are known to exhibit generally effective defence mechanisms to protect their eggs against parasitoids, e.g., thick egg chorions and oothecae or protective structures (e.g., scales, setae, feces, silk, spumaline; [3]). The rate of parasitization on an egg may be reduced by parental care [3] or laying of the eggs in aggregation, where the protected eggs are in the middle [36] and below the surface layer [37]. Some species of the family Chrysomelidae deposit feces not only on their larvae but also on the surface of eggs as protection against parasitoids [38]. However, no host defence has been observed in eggs of *Oulema* species until now.

Here, a new host defence mechanism of one *Oulema* species from Czech localities is demonstrated. The eggs of all three studied *Oulema* species have a thin sticky layer that allows the eggs to stick to the leaves of grain and grass on which they are laid [39], but the eggs from some populations in the Czech Republic, which are significantly prevalent only in the species of *O. gallaeciana*, also have a strong sticky layer on their surface. A similar defence has also been observed in galls with a sticky surface, which is a condition evolutionarily derived from galls without a sticky layer [40]. In that case, the parasitoids attempting to lay their eggs into a developing insect in the gall stick to the gall’s sticky surface [41]. Similarly, in the wasp *A. flavipes*, three situations that can happen during the parasitization of the eggs with a dark sticky layer from the Czech locality were observed:
(1)The female adheres to the sticky layer and is unable to either parasitize or release herself. First, before parasitization, the female needs to examine the suitability of the host eggs with her antennae (25; on average, for 12 s, *n* = 19 (Samková, unpubl.)). During this behavior, the wasp can adhere to the surface of the host egg before laying the eggs. In this case, the dark sticky layer succeeds in protecting the specific host egg.(2)The wasp is able to parasitize the egg but cannot unstick herself from the egg surface. Both of these host defence situations could be considered interspecific because other host eggs in the vicinity are protected against parasitization by this particular *A. flavipes* female.(3)The wasp successfully parasitizes and leaves the host, but afterwards, she must clean herself. At first, this defence may seem ineffective, because the sticky layer does not protect the egg from parasitization. However, this third observed behavior could lead to specialization in the wasps with such ‘experience’, which might afterwards prefer eggs without the dark sticky layer (such as those of *O. duftschmidi* and *O. melanopus*). It is known that, the choice of a host is related to the individual behavior and previous experiences of the female; flexible females could thus respond to a changing environment [19]. However, we must be careful in interpreting this claim, because scenario of wasps specialisation to host without defense againts parasitation is only our idea and future experiments are needed for it.

## 5. Conclusions

In this study, we described a new host defence (dark sticky layer on the host eggs of *O. gallaeciana*) against egg parasitoid wasp *A. flavipes*. This was shown by the absence of significant proof of any differences in host preference between the wasps from the Czech localities, where the eggs of *O. gallaeciana* have a specific defence (dark sticky layer), and those from the German locality, where *O. gallaeciana* eggs have no known specific defence. However, the question remains whether biological control will be still as effective in localities with this host defence. We assume that the rate of parasitism can be reduced by the presence of eggs with the dark sticky layer, which often prevents the affected wasp from parasitizing other eggs. These claims, however, require the support of future field experiments.

## Figures and Tables

**Figure 1 insects-11-00175-f001:**
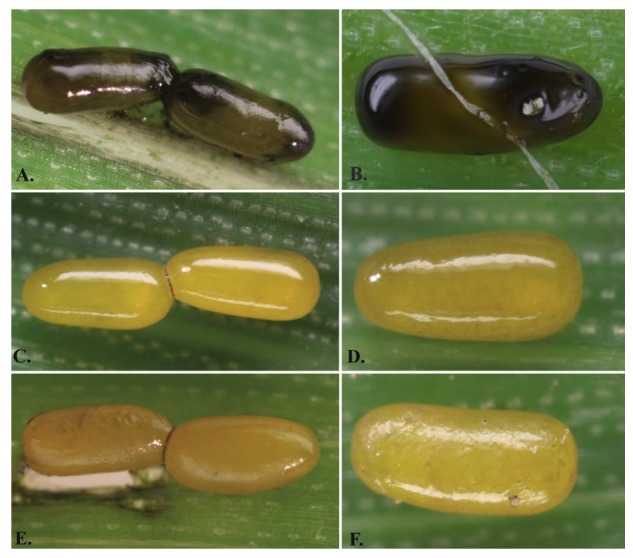
Three types of *Oulema* eggs: (**A**,**B**) sticky dark eggs (distinctive dark color; sticky layer); (**C**,**D**) little sticky eggs (yellow color; the structure of the egg surface is not visible under the sticky surface); (**E**,**F**) non-sticky eggs (yellow color; the structure of the egg surface is visible). Host eggs were not older than 24 h.

**Figure 2 insects-11-00175-f002:**
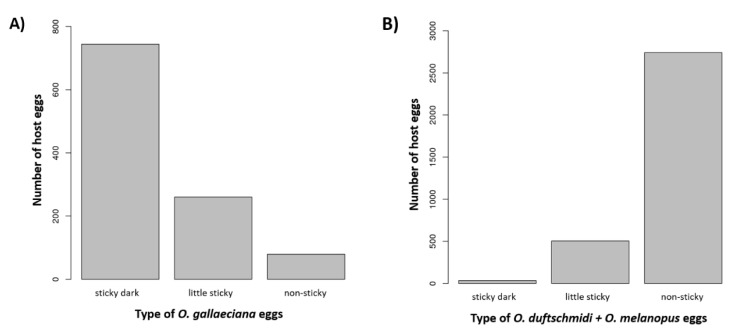
The graphs show the prevalence of three types of host eggs for *O. gallaeciana* (**A**) and *O. duftschmidi* + *O. melanopus* (**B**) from Czech Republic.

**Figure 3 insects-11-00175-f003:**
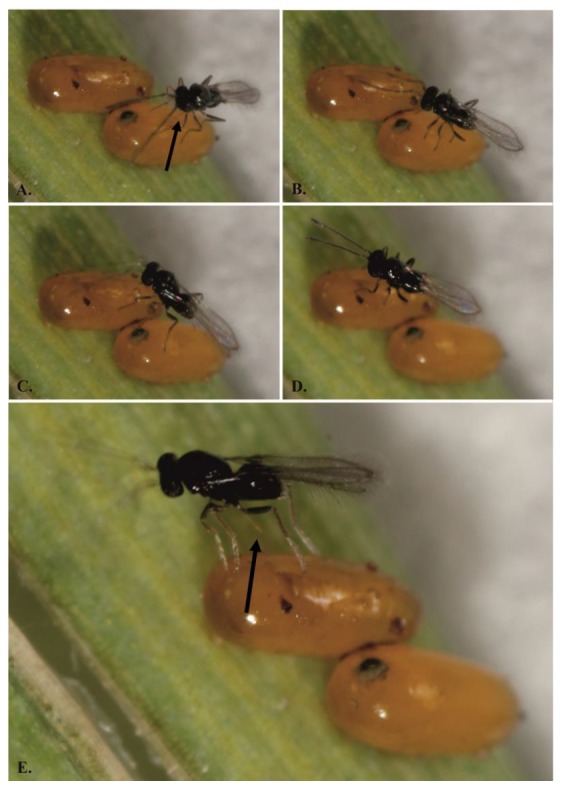
The parasitation on the non-sticky eggs by *A. flavipes*: (**A**) the female lays own eggs into the first egg of *Oulema* spp. (arrow indicates the ovipositor); (**B**) female ends the parasitization; (**C**) female examines the suitability of the second egg of *Oulema* spp. by its antennae; (**D**) the female lays own eggs; (**E**) female ends the parasitization (arrow indicates the ovipositor).

**Figure 4 insects-11-00175-f004:**
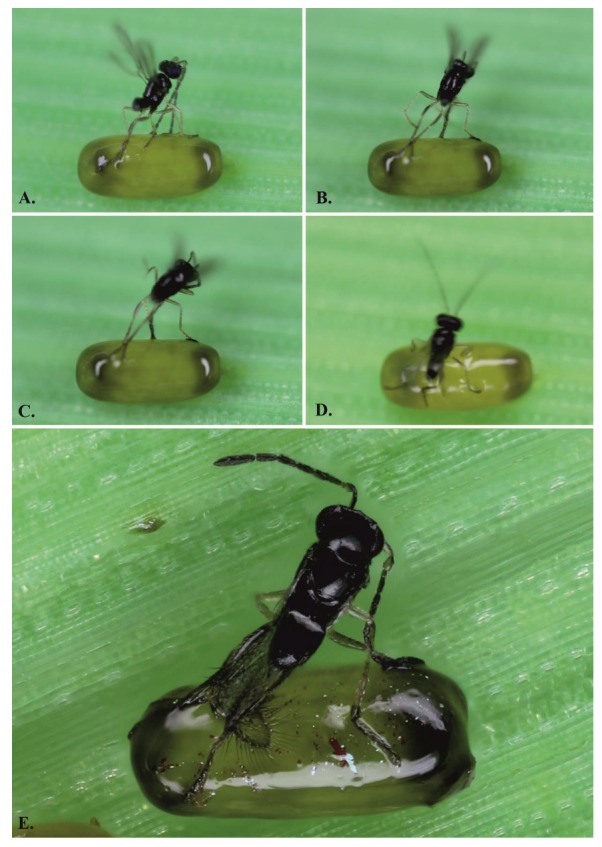
The parasitation on the host eggs with dark sticky layer by *A. flavipes*: (**A**–**C**) the dark sticky layer prevents the oviposition behavior of *A. flavipes*; (**D**,**E**) the wasps are unable to overcome the sticky layer of the host eggs.

**Figure 5 insects-11-00175-f005:**
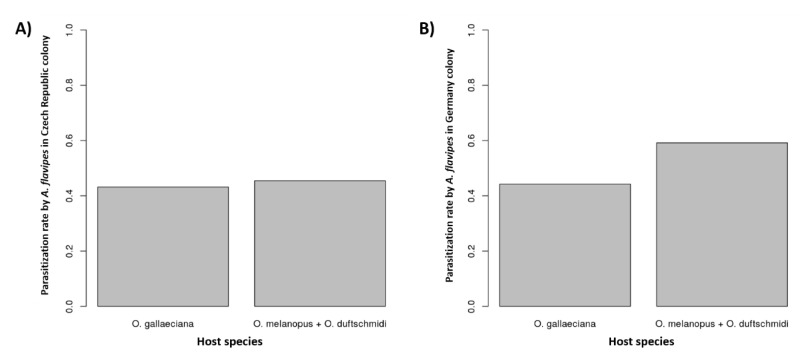
The graphs show the proportion of parasitized host eggs by *A. flavipes* wasps from localities, where the host defense (host eggs with dark sticky layer) was observed—Czech Republic (**A**), and the localities without host defense—Germany (**B**).

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
