# Peer review of "Host Specificity of the Parasitic Wasp Anaphes flavipes (Hymenoptera: Mymaridae) and a New Defence in Its Hosts (Coleoptera: Chrysomelidae: Oulema spp.)"

_insects, 2020, doi:10.3390/insects11030175_

Round 1

Reviewer 1 Report

Summary: 

The authors in this manuscript describe the host specific dynamics of the parasitoid wasp, Anaphes flavipes, across different populations of cereal leaf beetles which vary in their egg defensive capabilities. Certain populations of cereal leaf beetle display a sticky, black layer that coats this beetle’s brood, thus serving to either hinder or outright thwart parasitization from an adult female A. flavipes wasp. Their findings show that the central European parasitoid wasp, A. flavipes, does not exhibit host specificity, regardless of host defenses. The study and methods seem sound and have great potential. However, I have some minor concerns, mostly regarding their statistical analyses, which when clarified will be of great use to readers. Additionally, there are a few areas of grammatical, sentence structure, etc. to be addressed, which I have identified below - I hope that the authors/editors will be able to consult together to correct several of these issues across the manuscript prior to re-review.

Major Comments:

Lines 149-153 – The statistical tests performed need to be elaborated – what do the authors mean by “negative effect of the black sticky layer”? What is “Fisher’s method of meta-analysis”? Over-all, I am not entirely sure how to interpret the results presented, owing to lack of clarity in the statistical methods performed.

Additionally, none of the figures have captions, which make them extremely difficult for me to interpret, considering their reference in the manuscript.

Minor Comments:

Abstract:

Line 23 to 24 - introduce full species name for three Oulemabeetle species

Line 24 to 26 - “Recently...grammar/sentence structure; consider rephrasing

Line 26 to 27 - “locality...eggs” - grammar/sentence structure

Line 29 - “the host specificity” - phrase repeated twice in this line; rephrase to reduce redundancy 

Introduction:

More description 

Line 49 - parenthesis around “Girault, 1915”

Line 50 - parenthesis around “Say, 1832” and “Girault, 1928

Line 68 - “fairly common” could be rephrased to give more specific detail about Oulemaspecies abundance, relative to Lemaspecies

Line 70 - “leaves of cereals” - species name might be a good idea to include

Line 76 to 80 - consider rephrasing into two sentences

Materials and Methods:

In section header, add an ‘s’ to make “Material” plural

Needs more clarity in stats

More detail about German populations serving as negative control 

More description of how to ascertain different levels of 

Any data on host survival after parasitization

More details on measurements on successful emergence of host and wasp 

Line 84 - parentheses or commas should be put around “A. flavipes

Line 149 - “Fisher’s method of meta analysis”

Results:

Line 158 - “strongly significantly more prevalent” - grammar; consider rephrasing

Line 170 - “meta analysis of p-values from binomial tests” – not entirely sure what this means; please elaborate on the statistical analyses performed.

Discussion:

Line 174 - “In the past” - vague; specify how long ago?

Line 175 - grammar/phrasing

Line 215 - specify “female” refers to an adult A. flavipes individual

Line 217 to 218 - grammar/sentence structure

Supplemental Figure 2 - data from spreadsheet should be displayed to visually present host specificity, as measured by “the total number of parasitized host eggs of the three host species”; not clear what purpose these data sheets signify

Supplemental Figures 3 and 4 - clearly display the defensive, black coating on the host Oulemabeetles’ eggs; need captions for figures and labels

Author Response

REVIEWER: 1.

The authors in this manuscript describe the host specific dynamics of the parasitoid wasp, Anaphes flavipes, across different populations of cereal leaf beetles which vary in their egg defensive capabilities. Certain populations of cereal leaf beetle display a sticky, black layer that coats this beetle’s brood, thus serving to either hinder or outright thwart parasitization from an adult female A. flavipes wasp. Their findings show that the central European parasitoid wasp, A. flavipes, does not exhibit host specificity, regardless of host defenses. The study and methods seem sound and have great potential. However, I have some minor concerns, mostly regarding their statistical analyses, which when clarified will be of great use to readers. Additionally, there are a few areas of grammatical, sentence structure, etc. to be addressed, which I have identified below - I hope that the authors/editors will be able to consult together to correct several of these issues across the manuscript prior to re-review.

A (Authors): Thank you very much for all the suggestions to improve the manuscript. We agree, and we rewrote certain sections of the manuscript, especially methods and statistical data processing, and hope that we have better explained our experiments. We hope that all these corrections and changes have helped to improve our manuscript significantly.

Major Comments:

Lines 149-153 – The statistical tests performed need to be elaborated – what do the authors mean by “negative effect of the black sticky layer”? What is “Fisher’s method of meta-analysis”? Over-all, I am not entirely sure how to interpret the results presented, owing to lack of clarity in the statistical methods performed.

A: We rewrote these parts and we hope that now the manuscript is clearer and more suitable for Insects journal.

Additionally, none of the figures have captions, which make them extremely difficult for me to interpret, considering their reference in the manuscript.

A: We added it (it was our mistake, in a previous version of ms captions of figures were uploaded only into the submission application).

Minor Comments:

Abstract:

Line 23 to 24 - introduce full species name for three Oulema beetle species

A: Added.

Line 24 to 26 - “Recently...grammar/sentence structure; consider rephrasing

A: Changed.

Line 26 to 27 - “locality...eggs” - grammar/sentence structure

A: We believe that our English in this manuscript is correct. Before submitting, the manuscript was edited for proper English language, grammar, punctuation, spelling, and overall style by one or more of the highly qualified native English speaking editors at AJE (the certificate is attached).

Line 29 - “the host specificity” - phrase repeated twice in this line; rephrase to reduce redundancy 

A: Corrected.

Introduction:

More description 

Line 49 - parenthesis around “Girault, 1915”

A: Added.

Line 50 - parenthesis around “Say, 1832” and “Girault, 1928

A: Added.

Line 68 - “fairly common” could be rephrased to give more specific detail about Oulema species abundance, relative to Lema species

A: Changed.

Line 70 - “leaves of cereals” - species name might be a good idea to include

A: Added.

Line 76 to 80 - consider rephrasing into two sentences

A: Changed.

Materials and Methods:

In section header, add an ‘s’ to make “Material” plural

A: Added.

Needs more clarity in stats

 A: Corrected.

More detail about German populations serving as negative control 

A: Added.

More description of how to ascertain different levels of 

A: Added (it is shown in Fig. 1 Three types of Oulema eggs: A, B –sticky eggs distinctively dark color; sticky layer); C, D – little sticky eggs (yellow color; the structure of the egg surface is not visible under the sticky surface); E, F – non-sticky eggs (yellow color; the structure of the egg surface is visible).

Any data on host survival after parasitization

A: Added (The wasps A. flavipes is an egg parasitoid, when the host egg is parasitized, the larvae of the wasp will consume all the host egg and the host will not develop into the adult.).

More details on measurements on successful emergence of host and wasp 

A: Added (the parasitization by the wasp were recorded for each host egg).

Line 84 - parentheses or commas should be put around “A. flavipes

A: Added.

Line 149 - “Fisher’s method of meta analysis”

A: Yes, Fisher’s method of meta analysis. We added more details about the analyses.

Results:

Line 158 - “strongly significantly more prevalent” - grammar; consider rephrasing

A: Changed.

Line 170 - “meta analysis of p-values from binomial tests” – not entirely sure what this means; please elaborate on the statistical analyses performed.

A: Changed.

Discussion: 

Line 174 - “In the past” - vague; specify how long ago?

A: Changed.

Line 175 - grammar/phrasing

A: It is correct according to AJE.

Line 215 - specify “female” refers to an adult A. flavipes individual

A: Changed.

Line 217 to 218 - grammar/sentence structure

A: It is correct according to AJE.

Supplemental Figure 2 - data from spreadsheet should be displayed to visually present host specificity, as measured by “the total number of parasitized host eggs of the three host species”; not clear what purpose these data sheets signify

 A: We added the graphs for experiment “Host defense” (Suppl. Material 1) and “Host specifity” (Suppl. Material 2).

Supplemental Figures 3 and 4 - clearly display the defensive, black coating on the host Oulema beetles’ eggs; need captions for figures and labels

A: Added.

Reviewer 2 Report

The manuscript could provide a valuable contribution in defining the host composition and defense strategies adopted by Oulema species to contrast the attack of the parasitoid Anaphes flavipes. However there are numerous passages needed to improve the quality of this article.

First of all English needs considerable improvement. Second, in the introduction no information is provided about the literature associated to the biology of the parasitoid Anaphes flavipes and to the host spectrum susceptibility. Besides the Authors add no information if there are knowledge about similar or different defenses strategies in eggs of other host species.

The methods are poorly detailed it is sometimes unclear how experiment was set up (see some suggestion below)

The results are really reduced and limited to 15 lines. Consequently, the information provided is not clear.

 In line 139 Authors detail that host beetles Om+Od (after laying host eggs) were stored in 96 % ethanol and identified to species. But then in the result section the statistic is made on Om+Od and no separate information was added. 

 In the discussion part Authors refer to literature, but the information provided is not put in relation with the results achieved in the present research.

The conclusion suffers of the lacks in the previous part.

Some more detailed suggestions

Paragraph 2.1 this paragraph need to be completely reorganized

Line 87-88: which host was used in lab to obtain new parasites? please detail the procedure

line 88-89: where, which host was used in lab to obtain new parasites? please detail the procedure

line 91: it is asserted that each female was immediately mated but some question arises: where they mated? And where did they oviposit. “Parasitized eggs” is written just after oviposition (no logic relation is there.

Line 93: tube? Which tube? In the upper line it was written that parasitized egg were store individually in tubes.

Line 96. I think this part should be added before..

Paragraph. 2.2

Line 100: established  is not the most appropriate word

Line 111-112: major technical details need to be provided

Line 115-116: please change in: “eggs were removed on the 9th or 10th day after parasitization, placed in 115 1.5 ml Eppendorf tubes and stored at the same temperature in the thermal cabinet”

Line 123: singularly or in group? in line 125 Authors speak of single female oviposition? how could they separate? Please add information.

Line 131: please add information about the information acquired (give at least a list)

line 133- 135. it is difficult to understand the meaning of the phrase. Please rephrase.

Line 171. The information related to the absence of sticky egg layer was provided as an assumption for the control, consequently it is not a result.

Author Response

REVIEWER 2.

The manuscript could provide a valuable contribution in defining the host composition and defense strategies adopted by Oulema species to contrast the attack of the parasitoid Anaphes flavipes. However, there are numerous passages needed to improve the quality of this article.

A (Authors): Thank you very much for all your helpful comments listed below. We made many corrections and we hope that now the manuscript is clearer and more suitable for Insects journal.

First of all English needs considerable improvement.

A: We believe that our English in this manuscript is correct. Before submitting, the manuscript was edited for proper English language, grammar, punctuation, spelling, and overall style by one or more of the highly qualified native English speaking editors at AJE (the certificate is attached).

Second, in the introduction no information is provided about the literature associated to the biology of the parasitoid Anaphes flavipes and to the host spectrum susceptibility.

A: Added.

Besides the Authors add no information if there are knowledge about similar or different defenses strategies in eggs of other host species.

A: Added.

The methods are poorly detailed it is sometimes unclear how experiment was set up (see some suggestion below)

The results are really reduced and limited to 15 lines. Consequently, the information provided is not clear.

A: We rewrote the sections of materials, methods and results and added more information.

In line 139 Authors detail that host beetles Om+Od (after laying host eggs) were stored in 96 % ethanol and identified to species. But then in the result section the statistic is made on Om+Od and no separate information was added. 

A: Yes, all of the beetles were stored in 96% ethanol and they were assigned to species according to taxonomic characteristics, however, five specimens of O. duftschmidi or O. melanopus can not be identified to the species. Because, the O. duftschmidi and O. melanopus species are closely related and ecologically similar species and we did one category because of unequal number these species in the experiment.

In the discussion part Authors refer to literature, but the information provided is not put in relation with the results achieved in the present research.

A: The discussion has been changed to better put in relation published data and our results.

The conclusion suffers from the lacks in the previous part.

A: Added.

Some more detailed suggestions

Paragraph 2.1 this paragraph need to be completely reorganized

A: Changed.

Line 87-88: which host was used in lab to obtain new parasites? please detail the procedure

A: Added.

line 88-89: where, which host was used in lab to obtain new parasites? please detail the procedure

A: Changed.

line 91: it is asserted that each female was immediately mated but some question arises: where they mated? And where did they oviposit. “Parasitized eggs” is written just after oviposition (no logic relation is there.

A: Corrected.

Line 93: tube? Which tube? In the upper line it was written that parasitized egg were store individually in tubes.

A: Corrected.

Line 96. I think this part should be added before..

A: We rewrote this part.

Paragraph. 2.2

Line 100: established is not the most appropriate word

A: Changed.

Line 111-112: major technical details need to be provided

A: Added.

Line 115-116: please change in: “eggs were removed on the 9th or 10th day after parasitization, placed in 115 1.5 ml Eppendorf tubes and stored at the same temperature in the thermal cabinet”

A: Corrected.

Line 123: singularly or in group? in line 125 Authors speak of single female oviposition? how could they separate? Please add information.

A: Added.

Line 131: please add information about the information acquired (give at least a list)

A: All data are shown in Supplementary Materials 1 and 2. We have added 4 graphs showing the prevalence of host defense for O. gallaeciana, O. duftschmidi + O. melanopus host eggs and graphs showing the degree of parasitization for A. flavipes from Czech and German localities.

Line 133- 135. it is difficult to understand the meaning of the phrase. Please rephrase.

A: We hope that a shorter explanation (kept in manuscript) is better for understanding. Here we describe in more detail: Each mated female of A. flavipes (n = 59) had 12 host eggs available for parasitization for 8 hours in one Petri dish. As host eggs were used 4x Og with a sticky layer and 8x Od+Om without a sticky layer from localities in the Czech Republic (for illustration Supplementary material 3). This part of this experiment represented the finding of host specificity of the wasps from localities where a new host defense was observed as the black sticky layer on the host eggs Og.

Line 171. The information related to the absence of sticky egg layer was provided as an assumption for the control, consequently it is not a result.

A: Changed.

and without defense.

Reviewer 3 Report

In title: Change Oulema species to Oulema spp.

Insert comm after host (in its host, Oulema spp.)

I am assuming that the title does not require authority for species indicated for this journal-please double check

Line 18: insert comma after parasitic was and Anaphes

Line 19:  Change cereal leaf beetles to the cereal leaf beetle.

Lines 21-22: Why lack of knowledge about host specificity may have negative effect on the use of wasp? This is a very general statement that does not connect well with the next paragraph regarding experiments. Either specify what negative effects you are referring to or rephrase/delete this sentence.

Line 22: change carried out to conducted

Line 23: words “cereal leaf beetles” vs. “cereal leaf beetle” vs. “crop pests” appear often and interchangeably in the abstract. This usage is inconsistent and unnecessary. Although leaf beetles are a complex of species, they can be referred to as the cereal leaf beetle in context of it being an agricultural pest in cereals. My suggestion is to establish this fact early on and use a consistent term through out instead of referring to target organism with different terms

Line 26: specify which locality

Line 27: Again-specify what the control locality was. Were these two in same geographic region? Specifics here would help

Line 29-30: “the host specificity of the wasps from these two localities

was not confirmed” I am not sure if I understood this bit- do you mean that the wasp did not have specificity and irrespective of defense mechanism there was parasitization? Rephrase this and clearly state what the major finding was

Lines 30-32: This sentence is not clear. Also, abstract should clearly explain the nature of experiments conducted and results of this experiment and the implications of the study. Currently it is not clear as to what was done and how it helps derive the conclusion. There is still confusion between Oulema species and O. gallaeciana- was not later the focus of the experiment?

L31: Replace Oulema species with Oulema spp.

L41: replace “reduce” with “control”

L41-42: “Here the specific type of biocontrol….” Not sure what this sentence implies-rephrase

L43: Landscaping strategies? Do you mean landscape augmentation/diversification/modification strategies

L44: Replace conservative with conservation

L44-45: On the affected field? Consider rephrasing with “in the infested fields or areas”

L57-58: “and all organisms in a given habitat are potentially a host for parasitoids”: why ALL organisms in a given habitat have to be potential hosts? Do you mean that all organisms in a locality are potential hosts for one or other species of parasitoids? If that is what you imply, please rephrase.

L66: Insert comma between biological agent and Anaphes flavipes

L69: replace species with spp.

L72: delete in the past

L76-81: Authors describe new defense in Oulema eggs here for the first time. I hoped to see more description here on what the new defense was and what potential effects it may have on A. flavipes host finding and utilization. In absence of any such description it is difficult to establish the link between the need for experiments conducted and how they address the issue of improving the efficiency of A. flavipes as biocontrol agent. Because the main focus of this paper is the defense system in Oulema spp. the introduction would benefit from focused information on this defense system and how that may affect the efficacy of A. flavipes. This can then connect succinctly to the experiments conducted justifying their need.

L83: Replace parasitic wasps with Anaphes flavipes- Only one species was tested so no need to refer to them as parasitic wasps

L84: Delete parasitic wasps and start sentence with Anaphes flavipes. Also, replace in the periods from with between

L91: Were all males and females stored together and allowed to mate or were male:female pairs were maintained?

L95-96: 12 eggs of three species means 4 eggs of each species?

L102-103: Individual collection- do you mean hand collection?

L104-106: Not sure why the grouping was Od+Om: were these collected together and species determined only at the end of experiments based on genitalia?

L151: What do you mean by joining p-value? Are you referring to Fisher’s exact test?

L155: Statistical analyses don’t provide the details of the model tested. How many replicates were tested?

Fig. 2 and 3: Although the descriptions for the figures are present in the text, it would be of benefit to have the same description under the figures for better clarity

L174: Authors describe negative effects of parasitization: this should be explained upfront in the methods section somewhere explaining exactly what is a negative effect vs. what is favorable outcome etc.?

L223: The statement “The population of the wasp A. flavipes in central Europe is not host specific even in relation to host defence” overstimates the scope of current experiment. This involved localities only in part of Czech Republic and Germany and one experiment

Comments on results:

Are there no graphs/figures/tables showing actual data from the experiments- I did not see anything unless I missed any files. Is there only one experiment to test choice and make conclusions about whether the females chose the eggs with defense or not? Based on the current set up and in absence of actual data visualization, it is difficult to comment on rigor of experiments or to provide any further suggestions. The categories of preference have been broadly based on behavior. However, the data could have been graphically presented or at least tabulated. The overall results section is descriptive and it is in the results that reader’s get some clue about defenses and what categories of effects egg defense may have had on the behavior of the wasp. There is reference to chi-square test being conducted but details are lacking. Statistical analyses section significantly lacks clarity on metap package in R- this may be a commonly used package but information on analysis parameters and how the analysis was conducted must be provide to help assess the scientific rigor of the experiments. The choice of eggs in one experimental setup cannot be used to generalize whether the wasp has specific preference or not. What about wasps that cleaned themselves after effective rescue from eggs with sticky layers? Did they avoid freshly laid eggs with or without sticky layers? Were repeated tests to see learning process in female wasps observed? What is the effect on fitness of the progeny of the female host choice between eggs with defense and eggs without defense? Many of the questions can be probably answered using available data or reanalyzing some of these-or these may be presented differently to bring more clarity. Merits of the study and rigor can only be assessed if all the information is available.

Author Response

REVIEWER 3.

Thank you very much for reviewing our manuscript. We made many corrections and we hope that now the manuscript is clearer and more suitable for Insects journal.

In title: Change Oulema species to Oulema spp.

A (Authors): Thank you very much for your comment. Finaly, we decided to change the titlle as follows: Host specificity of the parasitic wasp Anaphes flavipes (Hymenoptera: Mymaridae) and a new defence in its hosts (Coleoptera: Chrysomelidae: Oulema spp.)

Insert comm after host (in its host, Oulema spp.)

A: See previous comment.

I am assuming that the title does not require authority for species indicated for this journal-please double check

A: Checked.

Line 18: insert comma after parasitic was and Anaphes

A: Added.

Line 19:  Change cereal leaf beetles to the cereal leaf beetle.

A: Corrected.

Lines 21-22: Why lack of knowledge about host specificity may have negative effect on the use of wasp? This is a very general statement that does not connect well with the next paragraph regarding experiments. Either specify what negative effects you are referring to or rephrase/delete this sentence.

A: Corrected.

Line 22: change carried out to conducted

A: Corrected.

Line 23: words “cereal leaf beetles” vs. “cereal leaf beetle” vs. “crop pests” appear often and interchangeably in the abstract. This usage is inconsistent and unnecessary. Although leaf beetles are a complex of species, they can be referred to as the cereal leaf beetle in context of it being an agricultural pest in cereals. My suggestion is to establish this fact early on and use a consistent term through out instead of referring to target organism with different terms

A: Corrected.

Line 26: specify which locality

A: Added.

Line 27: Again-specify what the control locality was. Were these two in same geographic region? Specifics here would help

A: Added.

Line 29-30: “the host specificity of the wasps from these two localities

was not confirmed” I am not sure if I understood this bit- do you mean that the wasp did not have specificity and irrespective of defense mechanism there was parasitization? Rephrase this and clearly state what the major finding was

A: Changed.

Lines 30-32: This sentence is not clear. Also, abstract should clearly explain the nature of experiments conducted and results of this experiment and the implications of the study. Currently it is not clear as to what was done and how it helps derive the conclusion. There is still confusion between Oulema species and O. gallaeciana- was not later the focus of the experiment?

A: Changed.

L31: Replace Oulema species with Oulema spp.

A: Changed.

L41: replace “reduce” with “control”

A: Changed.

L41-42: “Here the specific type of biocontrol….” Not sure what this sentence implies-rephrase

A: Removed.

L43: Landscaping strategies? Do you mean landscape augmentation / diversification / modification strategies

A: Added.

L44: Replace conservative with conservation

A: Corrected.

L44-45: On the affected field? Consider rephrasing with “in the infested fields or areas”

A: Corrected.

L57-58: “and all organisms in a given habitat are potentially a host for parasitoids”: why ALL organisms in a given habitat have to be potential hosts? Do you mean that all organisms in a locality are potential hosts for one or other species of parasitoids? If that is what you imply, please rephrase.

A: Changed.

L66: Insert comma between biological agent and Anaphes flavipes

A: Added.

L69: replace species with spp.

A: Changed.

L72: delete in the past

A: Removed.

L76-81: Authors describe new defense in Oulema eggs here for the first time. I hoped to see more description here on what the new defense was and what potential effects it may have on A. flavipes host finding and utilization. In absence of any such description it is difficult to establish the link between the need for experiments conducted and how they address the issue of improving the efficiency of A. flavipes as biocontrol agent. Because the main focus of this paper is the defense system in Oulema spp. the introduction would benefit from focused information on this defense system and how that may affect the efficacy of A. flavipes. This can then connect succinctly to the experiments conducted justifying their need.

We added a description of the figures. Especially, Fig. 1 shows three types of host eggs in relation to host defense. Fig. 3 shows host defense against a parasitic wasp, lines 215, 216 show the negative effect of host defense on the parasitization by a wasp. We added a graph with the prevalence of the type of egg for O. gallaeciana and O.melanopus + O. duftschmidi.

L83: Replace parasitic wasps with Anaphes flavipes- Only one species was tested so no need to refer to them as parasitic wasps

A: Corrected.

L84: Delete parasitic wasps and start sentence with Anaphes flavipes. Also, replace in the periods from with between

A: Corrected.

L91: Were all males and females stored together and allowed to mate or were male:female pairs were maintained?

We hope that a shorter explanation is better for understanding. Here we describe in more detail: Mating was carried out in different ways, but the wasps were randomly placed in the experiments and therefore we do not expect the effect of mating on the results of this experiments. The virgin females bred in Petri dish were mated by one or more males (several parasitized host eggs were in same Petri dish). The virgin females bred in Eppendorf tube were immediately mated by virgin males within the tube. If only the female(s) hatched from the parasitic host eggs, each female was placed into a Petri dish (5.5 cm in diameter) with a virgin male (not older than 24 hours post emergence) for two hours and mating observed.

L95-96: 12 eggs of three species means 4 eggs of each species?

A: Yes, we added.

L102-103: Individual collection- do you mean hand collection?

A: Yes, we changed.

L104-106: Not sure why the grouping was Od+Om: were these collected together and species determined only at the end of experiments based on genitalia?

A: Yes, it is described on the 125, 126 lines.

L151: What do you mean by joining p-value? Are you referring to Fisher’s exact test?

A: No, this Fisher’s method of meta analysis was used for joining the p-values (function “sumlog” in R package “metap” was used for this purpose) (more information https://en.wikipedia.org/wiki/Fisher%27s_method).

L155: Statistical analyses don’t provide the details of the model tested. How many replicates were tested?

A: We changed this part. All information about the analysis is explained in each experiment.

Fig. 2 and 3: Although the descriptions for the figures are present in the text, it would be of benefit to have the same description under the figures for better clarity

A: We added it (it was our mistake, in a previous version of ms captions of figures were uploaded only into the submission application).

L174: Authors describe negative effects of parasitization: this should be explained upfront in the methods section somewhere explaining exactly what is a negative effect vs. what is favorable outcome etc.?

A: We deduced the negative effect of the host defense against parasitoids (dark sticky layer) from the observed reproductive behaviour, which reduces the fitness of the wasp.

The reproductive behaviour of wasps is described in more detail in the results ((1) the female adheres to the sticky layer and is unable to either parasitize or release herself; (2) the wasp is able to parasitize the eggs but cannot release herself from the egg surface).

L223: The statement “The population of the wasp A. flavipes in central Europe is not host specific even in relation to host defence” overstimates the scope of current experiment. This involved localities only in part of Czech Republic and Germany and one experiment

A: Corrected.

Comments on results: 

Are there no graphs/figures/tables showing actual data from the experiments- I did not see anything unless I missed any files. Is there only one experiment to test choice and make conclusions about whether the females chose the eggs with defense or not? Based on the current set up and in absence of actual data visualization, it is difficult to comment on rigor of experiments or to provide any further suggestions. The categories of preference have been broadly based on behavior. However, the data could have been graphically presented or at least tabulated. The overall results section is descriptive and it is in the results that reader’s get some clue about defenses and what categories of effects egg defense may have had on the behavior of the wasp. There is reference to chi-square test being conducted but details are lacking. Statistical analyses section significantly lacks clarity on metap package in R- this may be a commonly used package but information on analysis parameters and how the analysis was conducted must be provide to help assess the scientific rigor of the experiments. The choice of eggs in one experimental setup cannot be used to generalize whether the wasp has specific preference or not. What about wasps that cleaned themselves after effective rescue from eggs with sticky layers? Did they avoid freshly laid eggs with or without sticky layers? Were repeated tests to see learning process in female wasps observed? What is the effect on fitness of the progeny of the female host choice between eggs with defense and eggs without defense? Many of the questions can be probably answered using available data or reanalyzing some of these-or these may be presented differently to bring more clarity. Merits of the study and rigor can only be assessed if all the information is available.

A: We added two graphs showing the prevalence of three host egg categories for O. gallaeciana and O. duftschmidi + O. melanopus from Czech and German localities and two graphs showing the host specificity of wasps A. flavipes, respectively the prevalence of parasitization of O. gallaeciana and O. duftschmidi + O. melanopus eggs by wasps from Czech and German localities.

These questions (What about wasps that cleaned themselves after effective rescue from eggs with sticky layers? Did they avoid freshly laid eggs with or without sticky layers? Were repeated tests to see the learning process in female wasps observed?) would require experiments with the constant observation of the wasps. In this work, Petri dishes with wasps and host eggs were stored in the climabox during the parasitization (8h).

To answer this question (what is the effect on the fitness of the progeny of the female host's choice between eggs with defense and eggs without defense?) we would need to do multi-generation experiments to find fitness (respective fertility) of offspring from host eggs with and without defense.

Round 2

Reviewer 2 Report

the Author improved the first version and added information  to the result session. 

Few typing mistakes are still present. 

evidenced in the attached file 

Author Response

Reviewer 2.

The Author improved the first version and added information  to the result session. 

Few typing mistakes are still present. Evidenced in the attached file 

A (Authors): Thank you very much for all your correction to improve the manuscript. We agree, and we corrected the manuscript.

Page 2

Comment 1

A: Added “of” (line 46)

Comment 2

A: Added “of” (line 47)

Comment 3

A: Replace “that” to “because” (line 88)

Page 3

Comment 1

A: Replaced “andthoseused” to “and those used” (line 105)

Comment 2

A: Replaced “hatched” to “emerged” (line 106)

Comment 3

A: Added “with” (line 124)

Page 4

A: Replaced “seeSupplmentary” to “see Supplementary” (line 106)

Page 5

Comment 1

A: Removed “The” (line 169)
